# Communication as a Part of Identity of Sustainable Subjects in Fashion

**Alena Kusá and Marianna Urmínová \*** 

Faculty of Mass Media Communication, University of Ss. Cyril and Methodius, 917 01 Trnava, Slovakia;
alena.kusa@ucm.sk
**\*** Correspondence: 2771312@student.ucm.sk; Tel.: +421-908-965-403

**Abstract:** Sustainability and corporate social responsibility have today become key assets of many successful businesses and corporations. Despite constantly growing environmental awareness, we are still facing the issue of overconsumption in both the textile and fashion industries. This is mainly due to improper marketing communication of sustainable subjects or a rather low level of consumers' awareness of sustainability issues. The main objective of the research is, through the opinions of selected authors and their studies, to compare the results of our own research focusing on Generations Y and Z and dealing with marketing communication of sustainable fashion. As part of the above objective, we attempt to propose the general trend in marketing communication of sustainable subjects. In order to reach this objective, we use the method of description and comparison of opinions of various authors, the analysis of the research questionnaire into the impact of marketing communication of sustainable fashion houses on consumers from Generations Y and Z and its comparison with previous research for the last three years. Thanks to the results of the research, we could observe that tools, media or forms of marketing communication of sustainable fashion producers have certainly changed. The research also provides answers to some of our questions in relation to the general interest of consumers in fashion sustainability, price as a decisive factor in the purchase of sustainable goods and the need for proper education in the field of sustainable fashion or any corresponding forms of marketing communication of sustainable subjects.

**Keywords:** consumer behaviour; marketing communication; sustainable fashion; corporate identity; education; Generation Y; Generation Z

---

## 1. Introduction

In its essence, fashion can be defined as a language by means of which a human's individuality can be expressed (Khandual and Pradhan 2018). The term sustainable fashion was first defined in the 1960s when consumers first observed the environmental impact of garment production and asked the fashion industry to change its production methods (Henninger et al. 2016). At first, ethical and ecological fashion was perceived rather negatively by consumers. The shift in thinking appeared after several industrial tragedies. According to Debbie and Danielle Moorhouse, it was the collapse of the fashion factory Rana Plaza in Bangladesh in 2013 which pointed to the issue of fast fashion as well as triggering the interest of consumers in ethical and fast fashion (Moorhouse and Moorhouse 2019). A total of 1130 people died and more than 2500 were badly injured. Not only this, but other tragedies have also not been sufficient stimuli to foster action in a sustainable manner unless people and consumers see the connection with their everyday behaviour.

The fashion industry has been the focus of researchers for many years. Fashion markets undergo fast changes and as a result, fashion businesses need to be flexible and prompt. This industry is known for its severe environmental impact. It entails an extremely long and complex suppliers' chain resulting

in excessive water and energy consumption, causing water, air or chemical pollution or generating large amounts of waste (Jacometti 2019). For this reason, the role of industrial sustainability seems to be much more important here than in any other field (Yang et al. 2017). Sustainability in the fashion industry, in respect of the seriousness of the current situation and taking into consideration all the above facts, is of the utmost importance and therefore, this matter needs to be dealt with on a regular basis and supported by up-to-date scientific findings and progress in this area. Consequently, we focus on the proper setting of environmental marketing communication of sustainable fashion producers and its effect on consumer behaviour of the members of Generations Y and Z. Consumer behaviour can only be changed through proper marketing communication of sustainable subjects combined with sustainability literacy of individuals.

Environmental marketing communication has now become an inseparable part of awareness in the field of sustainability and the fashion industry. Through their sustainable communication, businesses are trying to target the most individuals possible. Digital technologies enable them to act on both local and global levels. It is worth noting, however, that the transition to sustainable growth might require great changes to be made, but these are individuals who hold the power to change the whole world (Sherin 2013). Digital tools enable sustainable communication to be spread among individuals as well as consumers through different platforms—mass media, social media, company websites, blogs or applications.

Based upon the theoretical facts stated below, the objective of the study is to highlight the constant importance of the studied topic and as a matter of conclusion, propose the general trend for marketing communication of sustainable subjects. By comparing the opinions of the selected authors and their research with the results of our own study (2020), our goal was to observe the impact of environmental marketing communication and its forms on the perception of sustainable fashion by consumers from Generations Y and Z. In order to reach the key objective and the purpose of the study, at first it was necessary to determine the following research questions (RQ 1–5):

> *RQ 1 To what extent are the selected members of Generations Y and Z interested in sustainable fashion within their general interest in environmental issues?*

> *RQ 2 Is the price a decisive factor in the purchase of sustainable goods by consumers from Generations Y and Z?*

> *RQ 3 To what extent is fashion sustainability properly communicated by educational institutions?*

> *RQ 4 Are social media still regarded as a suitable form of communication of sustainable fashion?*

> *RQ 5 Is an entertaining form of communication of sustainable fashion still regarded as adequate?*

Answers to the above research questions were derived from the statements by various authors contained in the theoretical part of our work and presented by the results of the research questionnaire where the opinions and outcomes of the former research are discussed in more detail. Theoretical aspects primarily focus on fashion sustainability, consumer behaviour of Generations Y and Z and education as key factors within consumer behaviour and marketing communication of sustainable subjects. Theoretical aspects of the study serve as a basis for further observation of the proper setting of sustainable communication with the focus on Generations Y and Z.

## 2. Literature Review

### 2.1. Fashion Sustainability and Consumer Behaviour of Generations Y and Z

According to Soyer and Dittrich, a change in consumer behaviour leading to a higher level of activity in their environmental behaviour is the main precondition for improving fashion industry sustainability (Soyer and Dittrich 2020). Fashion is a kind of language by means of which a person's

individuality can be expressed (Khandual and Pradhan 2018). Consumption in various fields of the industry is influenced by a natural desire of a person to express his/her own opinions or identity. In general, consumers influence the fashion industry in various phases, i.e., through choice and availability of products, the number of selected products, preferences about product maintenance and their liquidation (Smith and Whitson 2018) or reusing and recycling methods respecting the principles of the circular economy. Promoting sustainable consumer behaviour requires knowledge of consumption indices in all their phases. Due to the role the identity and image represent to many consumers, those who wish to be "fashionable" overrun those who favour ethics and sustainability in their buying behaviour. This paradox highlights the conflict between the desire to consume on one hand and to limit one's consumption on the other (McNeill and Moore 2015). In their research, Birtwistle and Moore confirm that this is mainly due to insufficient knowledge of consumers vis-a-vis negative environmental impacts of the fashion industry (Birtwistle and Moore 2007). According to Emekci, consumers who do care about the environment should pay more attention to their consumer behaviour (Emekci 2019). This fact will be further discussed in Section 4.5—Summary of the results. In their study, Dabija and Băbuț also declare that, on one hand, businesses promote sustainable methods in order to gain a certain competitive advantage in markets defined by draconic regulatory measures, limited resources, climate changes or social pressure. On the other hand, there are consumers, mainly those from younger generations, who understand they need to change their approach and behaviour (Dabija and Băbuț 2019). The research we conducted focuses on consumers from Generations Y and Z. This generation is generally defined as a cohort group connected to a certain time period (Strauss and Howe 1991), known for its particular style of thinking or philosophy and raised under similar historical, social or cultural conditions. Generations Y and Z are two generations that were gradually led to environmental awareness due to growing pressure from businesses or individuals. However, a certain discrepancy can be observed here and that is the fact that the Millennials grew up with digital media and online technologies combined with green products and sustainability issues whereas Generation X became aware of these facts only at the time of their maturity (Euromonitor 2009; Oláh et al. 2019). Within our society, Generations Y and Z represent a strong economic factor because the evolution of their purchase behaviour may affect the environmental transformation of businesses including fashion producers.

## 2.2. Communication as a Part of the Identity of Sustainable Subjects

In contrast to individuals, relevant changes are currently being observed in regional, national or even multinational corporations including fashion brands. In their article, Alessandreo Da Giau and Laura Caniato et al., state that nowadays, fashion houses need to handle not only challenges that are caused by high market volatility, but furthermore, they have to adapt themselves to a newly developed business concept focusing on sustainability issues due to the higher sensitivity and awareness of consumers when it comes to ecological and social problems (Da Giau et al. 2016). Because of external pressure, fashion brands are forced to improve the sustainability of their activities and at the same time, continuously improve consumer awareness of where the garments come from, how they are produced or what their social and environmental impacts are (Verganti 2009).

Volatile fashion trends are forcing businesses to search for new ways of communication with the wider public in the field of ecological sustainability. From a managerial point of view, smaller retailers wishing to differentiate themselves from their competitors need to create a consistent company identity and inform their customers accordingly (Cheng et al. 2007). Shelby defines company communication as a set of communication forms and formats (Shelby 1993). It is noteworthy that company communication needs to cover an extensive scale of activities ranging from the company logo to managerial communication during public occasions up to promotion activities and product and service marketing (Karaosmanoglu and Melewar 2006). The authors Markwick and Fill confirm that marketing communication should be used to communicate the characteristic features of one's company identity (Markwick and Fill 1997). The main objectives of marketing communication are not

only placing products and services on the market, but also promoting the entire company. Company promotion and PR activities focusing on brand awareness, as well as particular products and services, enhance the company identity. Consumers' approach to company philosophy improves if these communication activities reflect the identity the company wishes to create in consumers' minds in the field of environmental marketing (Karaosmanoglu and Melewar 2006). Marketing activities, as defined by Choi and Sung, should mainly focus on consumer satisfaction, their social and ethical needs in the area of cultural support, environmental protection or disaster management (Choi and Sung 2013). Sustainable marketing activities help to improve company image, contribute to company growth and boost its vitality (Jung et al. 2020).

The greatest advantages of environmental marketing communication, as defined by Vilkaite—Vaitone and Skackauskiene, include: building relationships with customers, accruing profits, finding a way to reach business objectives, reinforcing competitive advantage, cutting costs or improving brand awareness (Vilkaite-Vaitone and Skackauskiene 2019). According to Lewandowska et al., creating effective and easy-to-understand eco-marketing communication is a real challenge (Lewandowska et al. 2017).

## 3. Materials and Methods

The main objective of our study is, through the opinions of the selected authors and their research, to compare the results of our own study (2020) focusing on Generations Y and Z and to look into marketing communication of sustainable fashion producers. As a part of our key objective, general trends for the marketing communication of sustainable subjects were proposed. In order to accomplish this objective, partial goals were determined.

The theoretical analysis focused on a content analysis of the statements of various authors from relevant scientific database studies and journals. The theoretical basis is supported by the description and comparison of opinions of authors in the field of sustainable fashion, environmental marketing communication and environmentally friendly consumer behaviour. The given theoretical knowledge was selected according to the content and time-relevance. The analytical part of the study consists of interpretation of results from the primary research carried out in 2020 which focused on the impact of marketing communication of sustainable fashion designers on Generations Y and Z. To evaluate the research, we used single statistical data analysis, frequency analysis, percentage evaluation, averages and a statistical method of semantic differential. The respondents were randomly selected from the above generations functioning in the online environment.

Each attribute of the main group had a non-zero probability of being included in the selected group. Since we do not know the exact total of the sample size, the size of the selected group was calculated by the following pattern:

$$n = \frac{Z_{1-\frac{a^2}{2}} \times \pi \times (1-\pi)}{E^2} \tag{1}$$

where: $\pi$ = the occurrence ratio of the observed trait within the basic set; E = maximum acceptable error interval; z = quantile of the distribution function.

As we do not know the occurrence ratio of the observed trait within the basic set, we conservatively determined the value as 0.5. The maximum acceptable error interval was determined at the level of 5% and the quantile of the distribution function had a value of 1.96, which equals 95 % reliability. The minimum size of the selected set according to this pattern is 385 respondents.

The research was carried out through an online questionnaire on a sample of 428 respondents from 27 August 2020 to 14 October 2020 during the coronavirus pandemic in Slovakia. When looking at the data in Table 1, we can see that especially representatives from Generations Y and Z took part in the research as these are more active in the online environment. The research focused on women

because generally speaking, they are, due to their consumer behaviour and fashion attitudes, regarded as the main target audience.

**Table 1.** Respondents' demographic data.

|  | **Category** | **N** | **%** |
|---|---|---|---|
| Gender | Men | 76 | 17.8 |
|  | Women | 352 | 82.2 |
| Age | 18–26 | 339 | 79.2 |
|  | 27–40 | 76 | 17.8 |
|  | 41–62 | 12 | 2.8 |
|  | 63+ | 1 | 0.2 |
|  | N = 428 | Total = 100.0% | |

Through the analysis, we focused on the basic demographic data of the respondents, their attitudes and approach to the environment as well as on what role price plays in the purchase of fashion products and their perception of environmental marketing communication in the fashion industry. The respondents' answers defining their own attitudes towards sustainable fashion and corresponding marketing communication form the basis for determining the general innovative concept of approaches in the given field. The basis of our previous research was to compare studies by the selected authors in the field of marketing communication of sustainable fashion. The significance of our study lies in the final deduction of the results of the research and their comparison with other studies. In addition, the study points to progress or prospective stagnation in the field of perception of marketing communication of sustainable fashion brands by their consumers over the course of the last three years.

## 4. Results and Discussion

After demographically defining the respondents of the research questionnaire, the following chapters will help us interpret the results of the research while providing answers to our research questions:

1. Respondents' interest in the environment and fashion sustainability
2. The main causes of consumers' disinterest in sustainable fashion products
3. Price as a decisive factor in the purchase of sustainable fashion garments
4. Marketing communication of sustainable fashion houses

*4.1. Respondents' Interest in the Environment and Fashion Sustainability*

The environment and its protection cover various problematic areas. Within our research (Table 2), we focused on the areas the respondents find the most important. Only 32.2% (138) of the respondents declare their interest in all environmental issues whereas 64.5% (276) of the respondents say their interest heavily depends on the specific area. Of the respondents, 3.7% (16) are not interested in the environment at all. As stated in Table 1, 71.5% (306) of consumers are interested in the recycling of consumer goods, 50.9% (218) in eliminating plastic packaging and products and 33.9% (145) in sustainable fashion issues. Clothes and fashion are an everyday part of the lives of younger generations by means of which their own individuality can be expressed. Based upon our theoretical findings, we may assume that the fashion industry represents an enormous global problem for our environment and, in this respect, has even overrun any other industry. Yet only a few respondents are really interested in sustainable fashion. It is obvious that consumers' buying decisions happen to be rather irrational and are not always connected with their values. Table 3. will try to define what deters consumers from Generations Y and Z from purchasing sustainable fashion.

**Table 2.** Respondents' spheres of interest.

| Section | N | % |
|---|---|---|
| Recycling of consumer goods | 306 | 71.5 |
| Composting | 122 | 28.5 |
| Tourism and Transport | 114 | 26.6 |
| Elimination of waste generation | 138 | 32.2 |
| Elimination of plastic packing and products | 218 | 50.9 |
| Sustainable fashion | 145 | 33.9 |
| Shopping in unwrapped stories | 98 | 22.9 |
| Exclusion of meat products from the diet | 70 | 16.4 |
| Environmental problem related to the cultivation, import and sale of cut flowers | 29 | 6.8 |
| Different answers | 4 | 0.9 |
| I have not mentioned this possibility in the previous answer | 92 | 21.5 |

**Table 3.** The main causes of consumers' disinterest in sustainable fashion products.

| Cause | N | % |
|---|---|---|
| The high price of sustainable garments | 182 | 44.4 |
| Non-attractivity | 109 | 26.6 |
| Insufficient advertising and poor marketing communication of sustainable fashion | 105 | 25.6 |
| A lack of brick and mortar stores | 182 | 44.4 |
| A lack of swaps and similar events | 74 | 18 |
| I feel embarrassed about going to second-hand shops (due to other people's prejudices) | 22 | 5.4 |
| Easy availability of cheap and fast fashion products | 106 | 25.9 |
| Fast fashion is affordable | 90 | 22 |
| Insufficient information about sustainable fashion | 82 | 20 |
| A different answer | 32 | 7.8 |

*4.2. The Main Causes of Consumers' Disinterest in Sustainable Fashion Products*

Despite better education in the field of the sustainable fashion industry on the part of the activists, the opening of local sustainable businesses or availability of sustainable fashion products, the majority of consumers tend to prefer fast fashion products. The results in Table 3 present the main causes of consumers' disinterest in sustainable fashion products.

The respondents proportionally agreed that the main causes of their disinterest in sustainable garments were their high price, as stated by 44.4% of them (182) or lack of brick and mortar stores, equally mentioned by 44.4% (182). In light of the aforementioned, we may say that the price has always been a decisive factor in the purchase of a fashion product. Due to the higher prices of sustainable products, consumers have started to perceive them as kinds of elite or exclusive goods that they cannot afford with their income. Today's globalised fashion industry seems to be corrupt from the inside. Fast fashion has taught consumers to anticipate that T-shirts should not cost more than 10€ per piece whereas repetitive fashion collections and season sales have persuaded us to buy them several times a month. Many sustainable fashion brands sell all of their products exclusively online, which seems to be another obstacle in their purchase—the absence of brick and mortar stores. Because of their higher price, consumers would certainly welcome to seeing goods with their own eyes, touching them or trying them on before they finally make a buying decision. The fashion industry and retail, as defined by Horská, show that high-quality furniture, sophisticated tiles or special lighting make the consumers feel at home (Horská and Berčík 2014). Fritzell (2018) also declares that Generation Y finds sustainable clothes expensive and unattractive. This is to say that if it is necessary to succeed with sustainable advertising and garments, it is also important to change respondents' attitudes and perceptions. To accomplish this goal, sustainable marketing should be entertaining and modern and should demonstrate different segments of sustainable garments to their customers other than just "white" or "grey" T-shirts. As the price is regarded as a decisive factor in the purchase of sustainable

fashion goods, taking into consideration living standards and the average wage in the Slovak Republic, this factor will be further analysed in Section 4.3.

### 4.3. The Price as a Decisive Factor in the Purchase of Sustainable Fashion Garments

The price of products and services is still considered the main and decisive factor for consumer behaviour. Aslam et al. state that the price highly correlates with consumer satisfaction (Aslam and Reema 2018). This is all defined in the price policy of sustainable products which are known to be much more expensive due to their production process, quality or the way they are liquidated. Vehmas et al. who based their study upon interviews (five respondents), research through the online platform Owela (50 respondents) and workshops with internal and external project partners (33 respondents), declare that price is an ongoing and decisive factor of sustainability when deciding to purchase a specific product (Vehmas et al. 2017). In the last three years since the above research, we must admit that consumer behaviour has not changed that much. Over 31.8% of respondents keep looking for good bargains in fast-fashion chains. When questioned about the connection between price and quality, consumers prefer branded products such as Adidas, Nike, and Vans, which, despite their high prices, are known for their ethical and ecological production or primarily sustainable materials used in the production process. Table 4 shows that the last places are occupied by consumers preferring cheaper second-hand garments and their concept is based on reusing the clothes that were thrown away and reintroduced into the market cycle. Only 17.8% of the respondents are willing to pay a higher price for sustainable fashion products which are produced under fair trade or environmentally friendly working conditions. The research shows that the environmentally friendly behaviour of the members of Generations Z and Y still heavily depends on the price of the selected products. As stated above, the price as a decisive factor is limited by the borders of the Slovak Republic and the corresponding economic conditions. The amount of the minimum and average wage under our conditions makes the price a decisive factor in the purchase of sustainable goods by Generations Y and Z. This assumption predicts further research in this respect.

**Table 4.** The price as a decisive factor.

| Factor | Answer | N | % |
|---|---|---|---|
| **Price** | I often buy cheaper garments at large fast fashion houses. | 136 | 31.8 |
| | I prefer cheaper second-hand garments. | 77 | 18 |
| | The price is not important to me as I buy what I like. | 33 | 7.7 |
| | No, I don't mind paying more for branded clothes (Adidas, Nike, Vans, etc.) | 79 | 18.5 |
| | No, I don't mind paying more for high-quality garments that are produced in fair trade and environmentally friendly conditions. | 76 | 17.8 |
| | Others | 27 | 6.3 |

### 4.4. Marketing Communication of Sustainable Fashion Houses

In the beginning, we focused on consumer preferences in connection with environmental issues while analysing the main reasons deterring the selected respondents from purchasing sustainable products. This will certainly enable us to properly determine the final recommendations while providing room for further research into this field. Other chapters of our paper will mainly deal with marketing communication of sustainable fashion goods which, despite the above-mentioned obstacles, may considerably enhance environmental awareness and eventually encourage consumers to purchase sustainable products.

In order to promote sustainable products (not only fashion), businesses have had to modify their marketing strategy to better communicate products' ecological features. Brands and businesses have started to use the new terminology, such as "eco, green, natural, organic, sustainable, etc."

Nevertheless, these promotional messages do not always bear an explicit definition, i.e., they lack accuracy and a consumer does not always have proper information about the materials, production methods and processes that were implemented, etc. (Yan et al. 2012). According to Debbie and Daniel Moorhouse, education is one of the best methods to attain evolution and promote changes across the whole fashion industry (Moorhouse and Moorhouse 2017). Therefore it is important to note that the proper marketing communication of sustainable businesses highly correlates with enhanced environmental awareness of consumers. Communication of sustainability issues should be exercised through multiple communication channels, such as social media, websites, radio, offline communication tools, etc. Sustainable products need to be connected to a story that is likely to catch consumer credibility (Čábyová 2018). Krajčovič assumes that effectiveness and viability of the advertising are one of the main criteria affecting the selection of a specific type of a medium that should be applied to a particular situation (Krajčovič et al. 2015). In their research, Morgan and Birtwistle state that a lack of information about sustainable fashion is mainly caused by insufficient media coverage (Morgan and Birtwistle 2009). According to Han et al., the target audience and the most effective forms of transmission should first be defined in order to set a proper communication strategy (Han et al. 2017). We should bear in mind the following facts:

- the general perception of marketing communication of particular subjects concerning sustainable fashion by consumers,
- to what extent sustainable fashion should be communicated through the above communication tools/media,
- in what form sustainable fashion should be communicated,
- the need for higher interactivity among sustainable businesses and followers in the online environment.

Within our questionnaire research, we used the method of semantic differential to define the perception of marketing communication of sustainable fashion by consumers. Particular businesses were split into five categories: sustainable fashion businesses, mass media, social sites, websites/blogs and educational institutions. Figure 1 uses *1* to define communication to a small extent and *5* for communication to a large extent.

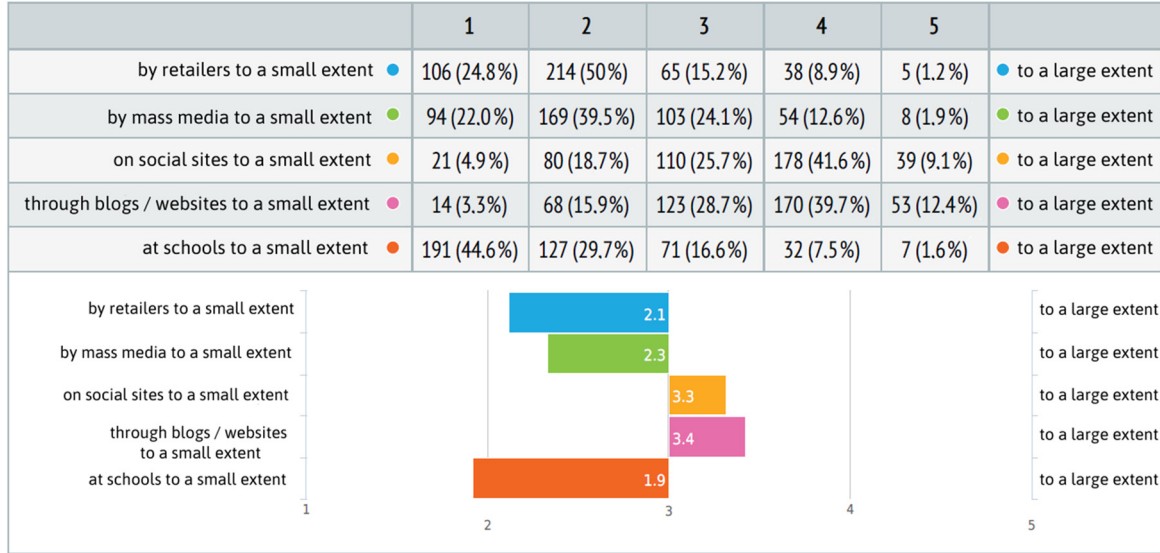

|  | 1 | 2 | 3 | 4 | 5 |  |
|---|---|---|---|---|---|---|
| by retailers to a small extent ● | 106 (24.8%) | 214 (50%) | 65 (15.2%) | 38 (8.9%) | 5 (1.2%) | ● to a large extent |
| by mass media to a small extent ● | 94 (22.0%) | 169 (39.5%) | 103 (24.1%) | 54 (12.6%) | 8 (1.9%) | ● to a large extent |
| on social sites to a small extent ● | 21 (4.9%) | 80 (18.7%) | 110 (25.7%) | 178 (41.6%) | 39 (9.1%) | ● to a large extent |
| through blogs / websites to a small extent ● | 14 (3.3%) | 68 (15.9%) | 123 (28.7%) | 170 (39.7%) | 53 (12.4%) | ● to a large extent |
| at schools to a small extent ● | 191 (44.6%) | 127 (29.7%) | 71 (16.6%) | 32 (7.5%) | 7 (1.6%) | ● to a large extent |

**Figure 1.** Extent of communication of the selected sustainable fashion subjects.

Figure 1 clearly defines that the proper setting of marketing communication of sustainable subjects requires proper awareness and education of consumers even though we all understand this topic

is hardly ever communicated at schools or educational institutions. Insufficient communication on the part of retailers or mass media is observed by the respondents themselves. As declared by Yan et al., sellers should use clearer and more concise messages when promoting environmentally friendly products as this might considerably boost consumers' awareness and at the same time, make consumers want to change their attitude to a specific environmentally friendly brand (Yan et al. 2012). In his study, Vehmas et al. (2017) also confirm that insufficient knowledge of sustainable fashion is often due to poor media coverage. The results of the research show that with regard to media space, consumers have not reported any considerable change in communication of sustainability over the last three years. On the contrary, higher interactivity has been reported on social networking sites, websites or blogs. The company website of a fashion producer plays a key role in the whole process of communication. One of the main advantages of using a web channel is the fact that it enables company information about sustainability to be directly posted on the website and make it available to their customers (Da Giau et al. 2016). Alessandra Da Giau et al. (2016) points out that even if almost all businesses maintain their company website, it was not usually directly set up to communicate sustainability initiatives, but rather to support brand differentiation among consumers and promote e-shopping. Generally speaking, the graph points to a lack of communication efforts on the part of educational institutions. In our opinion, environmental education is the key to change. The graph shows a considerable absence of sustainability communication on the part of educational institutions and schools in Slovakia. Our statement is challenged by Emekci whose research declares that even if most consumers were aware of environmental issues, they would certainly not pay attention to everyday environmental challenges (Emekci 2019). The results in Figure 1 indicate that further research into the issue of sustainability education through schools and educational institutions is needed.

We applied the same method (a semantic differential) to observe to what extent sustainable fashion should be communicated through the above-mentioned communication channels/media. According to the results carried out by Strähle and Gräff (2017), the most essential tools of sustainable fashion include various digital platforms of social media, mobile apps, blogs or online communities. These communities can be defined as social innovations enabling transformation towards a more sustainable society and development.

According to the respondents and on the basis of the results of the research we carried out (Figure 2), social media are seen as the most effective communication tools for sustainable fashion. They are one of the most fundamental transformation influences of informational technologies for entrepreneurship, within one entity or beyond its borders. They have triggered a revolution in many different ways in which businesses relate to the market and the society, thus creating a new world of opportunities and challenges across all business aspects ranging from marketing to finance or human resources management (Sinan et al. 2013). These days, social media provide complex opportunities for direct communication between a sustainable subject and a prospective consumer. This environment contains various interactive tools reinforcing the relationship between the above subjects, building loyalty or boosting information literacy and most essentially—prospective consumers can obtain information directly from the source. Interactivity as a driving force and as a tool having a considerable impact on a prospective consumer is further described in Table 5. Other recommended communication tools involve websites/blogs and influence marketing. We reported a slight stagnation in respondents' perception through websites/blogs in contrast to the survey by Strähle and Gräff (2017). On the other hand, influencer marketing and celebrity marketing (promotion through famous personalities and celebrities) have both reported steady growth. According to Sudha and Sheena, influencer marketing represents the process of identification and activation of individuals targeting a specific audience or a medium for the purpose of boosting the reach of the campaign or product sale (Sudha and Sheena 2017). The fashion industry is propelled, more than any industry ever, by influencer marketing. Influencers, famous personalities or celebrities are considered the driving force in the fashion market. Despite a lack of expertise or education in the given field, they tend to influence the buying decisions of consumers through their own personal opinion or preferences and consumers often admire them as specialists in

the given field. Our research suggests that the respondents prefer social events, workshops, events or offline marketing communication, but to a much smaller extent.

**Table 5.** Recommended forms of interactivity of sustainable businesses in the online environment.

| Form | N | % |
|---|---|---|
| Polls in posts/stories | 157 | 36.8 |
| The possibility for followers to express themselves and ask questions | 176 | 41.2 |
| Active electronic communication (email, Facebook, Instagram etc.) | 292 | 68.4 |
| Using specific hashtags | 170 | 39.8 |
| Attractive competitions upon fulfilment of specific conditions | 119 | 0.7 |
| Life broadcast | 39 | 9.1 |
| One's suggestion | 6 | 1.4 |
| Online environment does not require higher interactivity among brands, websites, businesses and prospective customers | 11 | 2.6 |

Based upon the results defining the recommended extent of use of communication tools/media by the respondents in the communication of sustainable fashion (Figure 2), it may be assumed that Generations Y and Z are reachable mainly through social sites or in the online environment. They favour entertaining and interesting forms of reception of new content and they like to be actively engaged in its creation. This phenomenon has also been captured in Table 5 which defines the need for higher interactivity among subjects promoting fashion sustainability as well as among followers and prospective consumers. The respondents tend to prefer active internet communication on the part of sustainable subjects or the opportunity to be asked or ask questions. To a large extent, they prefer participating in content creation or having the opportunity to use their own hashtags in posts about a sustainable subject. Active communication on the part of sustainable fashion designers or answering respondents' queries may also contribute to enhanced awareness of consumers in this field. The results in Figure 2 showed which communication tools should be used to promote sustainable fashion to Generations Y and Z. The information and the form of its communication to a consumer is, nevertheless, a key part of each communication strategy (Table 6).

**Table 6.** Recommended forms of marketing communication of sustainable businesses in the online environment.

| Form | N | % |
|---|---|---|
| educative/informative | 75 | 17.5 |
| entertaining (videos, podcasts, etc.) | 129 | 30.1 |
| critical–argumentative (highlighting extensive global consequences and impacts of the fashion industry) | 27 | 6.3 |
| combination of the above forms | 194 | 45.3 |
| different answer | 3 | 0.7 |

In his study from 2018, Fritzell pointed to the fact that Generation Y was considered a lazy generation. Therefore, businesses that wish to deliver some messages or information to this generation need to do so in an amusing manner because most users go online just to have fun. According to Fritzell (2018), social media users prefer videos because they require less energy and therefore, these have to be communicated in an amusing form to avoid rolling up the content. Within our research, 45.3% of the respondents prefer not only funny communication but also the combination of educative/informative, entertaining or shocking communication. When sustainable fashion is communicated, all the above forms need to be aligned. Education is of the utmost importance if we wish to motivate people to purchase sustainable garments. On the other hand, the young go online to search for fun and relaxation and therefore any form of education has to be creative and original.

Last but not least, it is necessary to highlight the disastrous impact of the fashion industry on the world, which can be effectively done only through shocking and critical advertising.

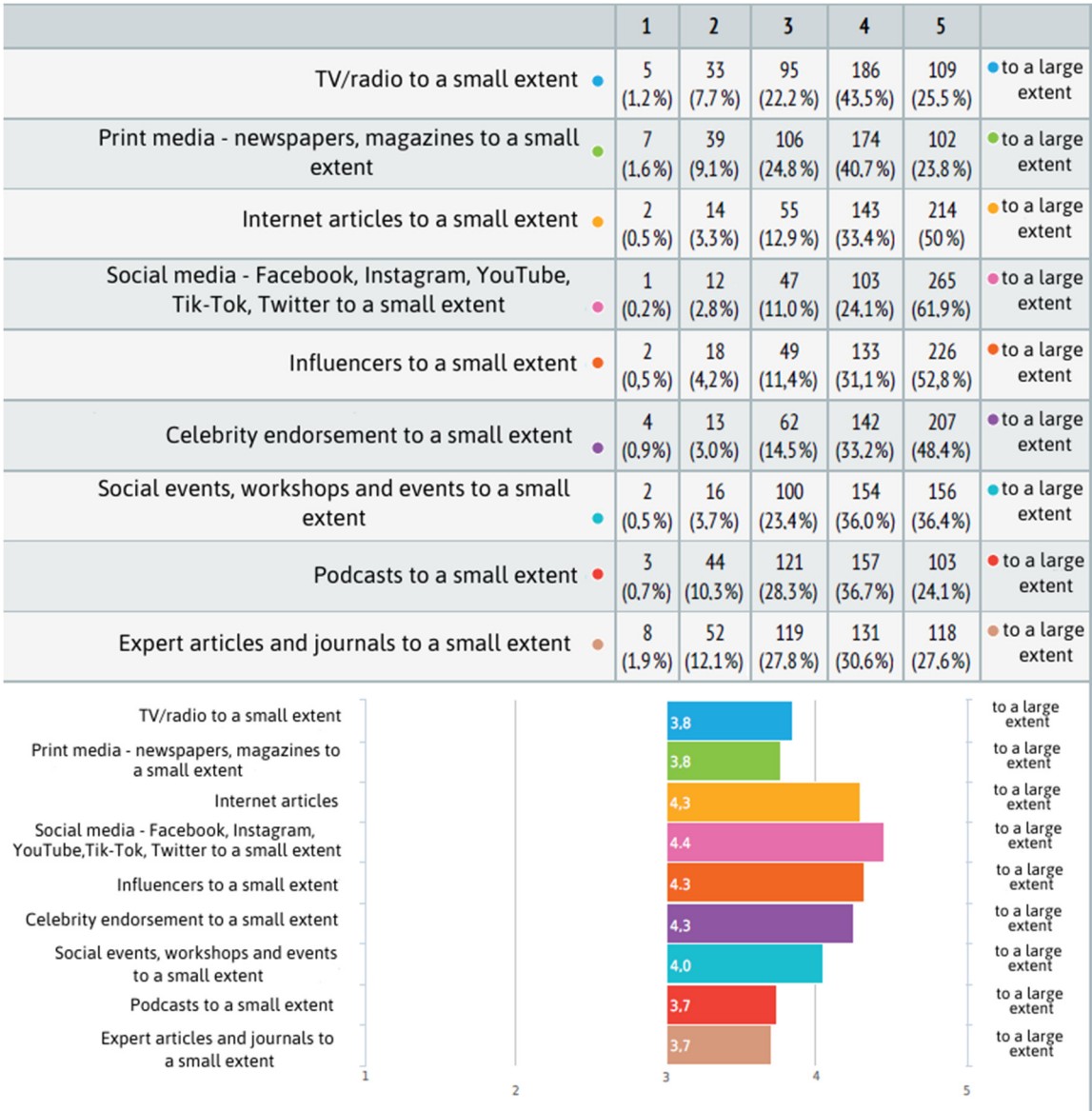

**Figure 2.** Recommended use of communication tools/media by the respondents in the communication of sustainable fashion.

### 4.5. Summary of the Results

In this chapter, we resume summarising the above research questions providing important conclusions and contributing to the given research objective. For the environment, Generations Y and Z are the main game changers but not only in the fashion industry. These changes and transformations of consumer behaviour, which substantially contribute to environmental pollution and corrupted morals and ethics in the fashion industry, cannot be attained without modifying the communication of sustainable subjects. Fashion and its communication strategy have always been a step ahead (Borboni 2019). According to Dabija et al. generation Z prefers to purchase products from companies that apply sustainability principles and attempt, by means of social campaigns (Dabija et al. 2020). "Communication of sustainability", as described by Morsing and Schultz (2006), can be defined as a set of strategies and subsequent processes which play a key role in communicating

information about the company's environmental and socially responsible behaviour with the aim of influencing, supporting and improving the company image to all the parties involved and end-users. Nowadays, every business, including fashion designers, should notify the public and consumers of its environmentally friendly and sustainable activities by combining various information media channels.

*RQ 1 To what extent are the selected members of Generations Y and Z interested in sustainable fashion within their general interest in environmental issues?*

Thanks to our research, we managed to specify not only the general sphere of interest of the respondents in sustainable fashion but also to set appropriate communication tools and marketing communication of sustainable subjects. Consumers' interest in environmental issues is mainly influenced by the long-term reception of information or pressure the central authorities exercise on districts or municipalities, such as recycling of consumer goods and elimination of plastic packaging and products. According to our results, sustainability in the fashion industry is the third most discussed sphere of interest of the respondents (33.9%) despite the fact that the fashion industry is the second most polluting business in the world.

*RQ 2 Is the price a decisive factor in the purchase of sustainable goods by consumers from Generations Y and Z?*

As shown in the research, the high price of sustainable fashion garments seems to be the one and only reason deterring consumers from buying these types of products. Price as a decisive factor was further observed by Vehmas et al. in their article named *Consumer attitudes and communication in circular fashion* based upon which we aimed to observe the state of the issue over the last three years. The analysed study had a qualitative character and therefore no quantitative information about consumer behaviour was obtained. The article methodics consisted of interviews (5 respondents), research through the online platform Owela (50 respondents) and workshops with internal and external project partners (33 respondents). One of the key aspects observed in the above interviews was the price the respondents regarded as a decisive factor of sustainability when deciding to purchase a specific product. We reported continuous stagnation compared to the results of our research involving 428 respondents. However, it is noteworthy that price as a decisive factor is more or less limited by the borders of the Slovak Republic and the economic conditions arising herein. Therefore, this finding cannot be generalised.

*RQ 3 To what extent is fashion sustainability properly communicated by educational institutions?*

Environmental awareness in the fashion industry, apart from the necessary education of consumers, may be accomplished through proper marketing communication of sustainable subjects.

The change in consumers' perception and behaviour goes hand in hand with their education in the field of fashion sustainability. Education provides unlimited space for starting the change we need to see in the world. Education is closely linked to the perception of the price of sustainable fashion goods. Consumers need to understand that the price of sustainable garments reflects ethically paid workforces, human and environmentally friendly production, quality, material, and marketing, etc.

*RQ 4 Are social media still regarded as a suitable form of communication of sustainable fashion?*

In their article "*The Role of Social Media for a Sustainable Consumption*", Strähle a Gräff primarily focused on social media. One of the key findings of the above research is defining eight roles of social media affecting sustainable consumption in contrast to former research in the given field. The most significant and beneficial chapter in our research deals with the use of various digital platforms of social media in order to attain effective communication. In 2017, when the study by Strähle and Gräff was carried out, the most powerful tools of social media included mobile applications, blogs, Instagram or swaps and communities sharing clothes or other goods. Marketing communication of sustainable

fashion brands targeting consumers from Generations Y and Z, as demonstrated by the results of our research, should also benefit from the space provided by social media, websites/blogs as well as promote cooperation with influencers or famous personalities. Influencers, famous personalities or celebrities are nowadays considered the most powerful agents in the fashion market despite lacking education in the given field. The proper interactive marketing communication of sustainable subjects should involve active communication through various communication tools provided by social media, such as the opportunity for followers or prospective customers to comment on questions or their active engagement in marketing campaigns by means of various hashtags (#secondhandislove, #sustainablefashion a pod.).

*RQ 5 Is an entertaining form of communication of sustainable fashion still regarded as adequate?*

Our studies have observed several fundamental shifts in consumer behaviour and respondents' preferences over the last three years. The research by Fritzell (2018) is regarded as a starting point, as the author focused on social media as a communication channel of fast fashion producers who tend to use social media for promoting their own sustainable products. The author used a qualitative observation method by means of semi-structured individual interviews with selected respondents of Generation Y. The study also defined a suitable form of communication of sustainable subjects to be applied to social media. The results of this qualitative research point to the fact that Generation Y is said to be a lazy generation and consequently, businesses which intend to communicate various messages and information to this generation have to do so in an amusing way as most users perceive social networks as a place for having fun. The research also defines a certain shift in the above field as the selected target group prefers not only an entertaining but also an educative/informative or even shocking form of communication. They do understand that this global ecological problem cannot be dealt with only amusingly, but, quite the contrary. Due to the seriousness of the situation, people need to be aware of the catastrophic consequences of their unconscious and comfortable lifestyle.

The results of our research suggest that sustainability should be described as a continuous concept that needs to be adapted to educative and marketing standards (Kong et al. 2016).

## 5. Conclusions

The issue of sustainability and sustainable fashion is highly trendy and stylish. The textile industry is one of the most polluting industries in the world. Many experts publishing environmentally friendly declarations and studies have already been supported by other entrepreneurs, fashion designers or even consumers.

In the light of the unfavourable situation in the fashion industry we might observe these days, this paper is trying to support sustainable fashion subjects and through the answers to our research questions, it also presents general communication trends of these brands. It provides answers when it comes to the interest level of Generations Y and Z in sustainable fashion within their general knowledge of environmental issues. It appeals to the price as a decisive factor or specifies effective tools and forms of effective marketing communication. It also assumes how important it is to enhance environmental literacy and sustainability awareness, which are the main predispositions for any behavioural changes in our future generations. Representatives or managers of sustainable subjects should try to use the wide spaces provided by social media for improving communication and interactivity with their prospective customers. They should not act as mere sellers of sustainable goods but as active promoters of environmental awareness in their given fields. Their communication should contain the attributes of entertainment, education or even fear marked by the seriousness of the situation.

As mentioned above, the study was also limited by other factors. Due to the COVID-19 pandemic and subsequent restrictions, the research was limited only to the territory of the Slovak Republic and could not be conducted in any other European or overseas countries. The research questionnaire was distributed online to two key generations of consumers Y and Z. To make the research more significant and applicable to other areas, it would be interesting to focus on Generation X, Baby Boomers or the

Generation of the Connected-C. The presented research has also given us the prospects for a future study of the decisive factors playing a key role in the purchase of sustainable fashion goods—price and education.

In today's globalised society defined by hyper-competition in the fashion industry, it is utterly indispensable to think over all processes within the circular economy, the ethical issues of textile production, fair trade, reduction of carbon footprint and many other economic and ecological aspects affecting our planet and all humanity.

**Author Contributions:** Conceptualization, M.U. and A.K.; methodology, A.K.; software, M.U.; validation, M.U.; formal analysis, M.U.; investigation, M.U.; resources, M.U.; data curation, M.U.; writing—original draft preparation, M.U.; writing—review and editing, M.U.; visualization, M.U.; supervision, A.K.; project administration, M.U.; funding acquisition, A.K. All authors have read and agreed to the published version of the manuscript.

**Funding:** This research was funded by project VEGA 1/0078/18 Aspects of marketing communication in the management processes of the circular economy.

**Acknowledgments:** This contribution is a partial result of the project VEGA 1/0078/18 Aspects of marketing communication in the management processes of the circular economy.

**Conflicts of Interest:** The authors declare no conflict of interest.

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
