# Peer review of "Communication as a Part of Identity of Sustainable Subjects in Fashion"

_jrfm, doi:10.3390/jrfm13120305_

Round 1
Reviewer 1 Report
Dear Author(s),
This is an interesting paper on a topical research area. While the topic is clearly aligned with our journal’s aim and scope, there are several concerns regarding the practical relevance and conceptual foundation for your study, as well as the research design, that make it difficult to determine if or how your study makes a significant contribution to previous research in this area.
It is recommended that the authors address the following comments and suggestions.
ABSTRACT
Please add a clear research objective in the abstract. As this should be the starting sentence for the abstract, you have to try to sharpen its focus.
Overall, I would make the abstract sharper, answering to the following questions:
- What is the problem?
- What have you done?
- What is the main contribution of the paper?
INTRODUCTION
the introduction section is quite confusing. Please rewrite it following the items below:
- Establish the importance of research.
- Establish a theory based gap.
- Explain contribution.
- Present the overall paper structure.
LETERATURE REVIEW
Your introduction and literature review should be like a story to read. This means that they should be interrelated: the introduction presents the main topics, then they have to be explored in deep in the literature review. This is not the case.
Moreover, in the whole paper, the authors used some old citations and it is suggested to read the recent papers and add citations.
METHODOLOGY
Methodology section mainly is the paper’s argument built on an appropriate base of theory, concepts, or other ideas.
DISCUSSION
There is a need for more discussion on how achieved findings can be connected to the previous conceptual background. Please link the discussion of the research problem with the highlighted gaps in the conceptual background.
Please, make some linkage between the paragraphs.
There are some typos, please make sure you proofread the paper
THEORETICAL AND PRACTICAL IMPLICATIONS
To begin, the justification for your study from an industry/practical standpoint was lacking. This was a salient oversight on your part.
You should try to answer to the following questions: What kinds of objective evidence can you offer that would make industry leaders sit up and pay attention to your study? What makes this topic a big deal right now, and perhaps in the immediate future?
Answers to these and related questions will help make a much stronger case for pursuing this line of inquiry.
Additionally, the conceptual foundation for your study was underdeveloped. In particular, you did not clearly articulate how your study advances that which we already know. Indeed, it appears that previous research supports each of your hypotheses. Thus, it's not evident how the current study amends or complements previous work.
CONCLUSIONS
I encourage you to clearly and cogently explain final section: (1) how the focal study addresses an important priority; (2) how the topic is connected to existing theory; (3) what we already know about practice and theory; (4) what specifically we do not know; (5) why we need to know what we do not know; and (6) how this study or inquiry will help close the practical and theoretical gaps between what we know and do not know.
Best Regards
Author Response
Dear Reviewer,
In the first place I would like to thank you very much for your comments, which
moved our study forward. We have taken a note of your comments and we have supplemented,modified and have rewritten the study.
First of all, we integrated the uniform main goal in individual parts of the study. We have extensively reworked the introduction of our work, where we supplemented the meaning and the need of the article, we described the structure and we determined the research questions that form the basis of the analytical part of the article.
The literature review was supplemented with newer publications reformulated, logically arranged and we added by the issue of generations.
Within the methodology of work, we have again unified the goal of our work.
The chapter results and discussion was completely logically re arranged.
Individual partial results were logically interconnected and supplemented with a critical evaluation of the findings. Chapter 4 has been renamed to Chapter 3.5 Summary of the results and it clearly describes the results of our survey based on predetermined research questions.
The conclusion of the paper points out the benefits of the article, describes the managerial implications that managers should learn from our work, describes the limitations of the work and sets out possible perspectives for further research.
The English language was re-evaluated and checked twice in the work.
We believe that it is all right.
All the best in your business,
Faithfully,
Mgr. Marianna Urmínová
Prof. Ing. Alena Kusá, PhD.
Reviewer 2 Report
Thank you for submitting your paper. The article provides an interesting analysis of the factors that support or limit the development of sustainable business in the fashion industry. The scientific relevance of this paper is original and important. However, there are many shortcomings.
Comments on individual parts:
- The title of the article is correct, however, it could be improved.
- The abstract is not precise enough and it is different from the objective included in the Introduction section. Some sentences require improvements.
- The Introduction highlights the research problem and defines the research gaps based on utilizing a current literature review. However, the research objective is not precise. It would be important to consider analyzing factors that support or constrain sustainable business development in the fashion industry. The research questions or hypotheses are missing.
Moreover, the introduction should provide not only a research background and research objective but also a brief indication of the methods used and a brief description of the content of each section of the paper (in the last paragraph).
- The use of research methodology is adequate. It is difficult to understand why the author defines in the first sentence of this section a different research goal than in the Introduction and Abstract. What is actually a research goal? The methods used should be better justified. How was the data analysis performed?
- The results section contains a broad data analysis, which is only partially reflected in the research purpose of the article (e.g. the findings included in subsection 3.2 and 3.4 are not reflected in the research objective). The analysis of the results was conducted in a simple way, which significantly reduces the quality of the results. The result section should include not only data presentation but also a critical assessment of the findings.
- The discussion part should be better structured by referring to the key findings. It should embrace an extensive explanation of differences and similarities among the findings obtained and those of other scholars. The concept presented in Table 6 is not clear and poor in terms of scientific quality. What is the theoretical contribution of this study?
- There is a lack of a conclusion section. The conclusions section should include a synthetic overview of the key research results. The authors should also indicate practical or/and theoretical implications, research limitations, and potential directions for further research.
The academic language is correct. However, general proofreading would be advisable.
Author Response

(The authors gave the same response as above.)

Reviewer 3 Report
Thank you very much for allowing me to review this very interesting paper. I enjoyed reading the paper. I think it is well written and properly argued. However there are some minor aspects that need to be fixed / should be addressed by the authors.
It would be good to make a connection between the research scope of the paper and a theory which you try to enhance. As you speak of Gen Y and Z it would be proper to also consider the generational theory, like in this paper:
https://www.mdpi.com/2071-1050/11/17/4532
https://onlinelibrary.wiley.com/doi/abs/10.1002/cb.157
https://www.sciencedirect.com/science/article/abs/pii/S096969891200149X?via%3Dihub
https://www.sciencedirect.com/science/article/abs/pii/S0969698915300874?via%3Dihub
Abstract
" The main objective of this study is " you mean probably the research scope of the research question
Introduction
"and ecological fashion was " i think you mean organic fashion ... the literature also speaks of green fashion.
"Debbie and Danielle Moorhouse, it was the " reference for that?
The introduction should clearly stress out the research gap, the research question and explain how the paper is linked to an existing theory. Furthermore the paper should also explain here how the research question is implemented in the different sections of the paper, especially in the methodology. It should be clear how the paper is original and novel for the international literature. The last paragraph should briefly present the next sections of the paper.
I am missing the lit review on the key concepts.
I would recommend that the introduction is divided from the literature review. The literature review should be focused on the key concepts on that the paper is based.
The methodology should be further developed, by stating what method the authors have used, what research instrument(s), how they have collected and analyzed data an in which order.
If you have an online questionnaire than you have different statements. These statements should been taken from the international literature, i.e. based on scales. So please present the operationalisation of those scales. Please also state something about the reliability, validity and internal consistency of the data you have collected.
Section 3 is entitled "results and discussions" and section 4 "discussions". You should either merge the 2 or discard "discussions "from section 3.
The presentation of results and the discussions are well developed.
Conclusions need to be enhanced and enlarged. You need to have 4 paragraphs:
a) theoretical implications: how does the paper contribute originally in extending the literature. i.e. the generational theory
b) managerial implications: what are the 2 or 3 main aspects that managers should learn from your paper.
c) limitations of own results
d) future research perspectives.
I think that some more papers from the last 3 years should be cited.
Author Response

(The authors gave the same response as above.)

Round 2
Reviewer 1 Report
Dear authors,
I'm happy with the changes you have made to the paper.
Best Regards
Author Response
Dear Reviewer,
I appreciate your answer. Thank you very much.
All the best in your business,
Faithfully,
Mgr. Marianna Urmínová
Prof. Ing. Alena Kusá, PhD.
Reviewer 2 Report
Thank you very much for submitting the revised version of your paper based on the reviewers' comments.
In general, this paper can be accepted for publication after improving the following shortcomings, e.g.:
- the numbering of section 2.1 on page 3 is incorrect - it should be 2.2.
- it would be important to explain how the reliability and validity of the collected data was ensured
- Table 1. Respondents’ demographic data should be included in section 2
- there is no clear discussion of results, how does your study develop what we already know? To what extent does your study confirm or expand on previous research?
Author Response
Dear Reviewer,
Thank you very much for your answer.
We appreciate your comments.
In our study, we changed the numbering of sections. We explained the reliability and validity of the collected data (Materials and Methods ). Table 1. - Respondents’ demographic data was included in section 2.We have expanded chapter 4.5 - Summary of the results a discussion of the extent to which the study confirms or extends previous research.
We believe that it is all right.
All the best in your business,
Faithfully,
Mgr. Marianna Urmínová
Prof. Ing. Alena Kusá, PhD.